# Automatic Flight Callsign Identification on a Controller Working Position: Real-Time Simulation and Analysis of Operational Recordings

Raquel García [1,*], Juan Albarrán [2], Adrián Fabio [1], Fernando Celorrio [1,2], Carlos Pinto de Oliveira [3] and Cristina Bárcena [2]

1   Centro de Referencia I+D+i ATM (CRIDA A.I.E), 28022 Madrid, Spain; afabio@e-crida.enaire.es (A.F.); fcelorrio@enaire.es (F.C.)
2   ENAIRE, 28022 Madrid, Spain; jaalbarran@enaire.es (J.A.); cpbarcena@enaire.es (C.B.)
3   EML Speech Technology GmbH, 69120 Heidelberg, Germany; carlos.pintodeoliveira@eml.org
*   Correspondence: rglasheras@e-crida.enaire.es

**Abstract:** In the air traffic management (ATM) environment, air traffic controllers (ATCos) and flight crews, (FCs) communicate via voice to exchange different types of data such as commands, readbacks (confirmation of reception of the command) and information related to the air traffic environment. Speech recognition can be used in these voice exchanges to support ATCos in their work; each time a flight identification or callsign is mentioned by the controller or the pilot, the flight is recognised through automatic speech recognition (ASR) and the callsign is highlighted on the ATCo screen to increase their situational awareness and safety. This paper presents the work that is being performed within SESAR2020-founded solution PJ.10-W2-96 ASR in callsign recognition via voice by Enaire, Indra, and Crida using ASR models developed jointly by EML Speech Technology GmbH (EML) and Crida. The paper describes the ATCo speech environment and presents the main requirements impacting the design, the implementation performed, and the outcomes obtained using real operation communications and real-time simulations. The findings indicate a way forward incorporating partial recognition of callsigns and enriching the phonetization of company names to improve the recognition rates, currently set at 84–87% for controllers and 49–67% for flight crew.

**Keywords:** speech recognition; human–computer interaction; situational awareness; air traffic management; air traffic controller; flight callsign; ASR; VRS

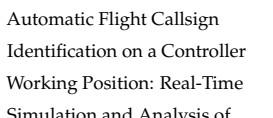



## 1. Introduction

ATCos work with a Controller Working Position (CWP) which displays all of the information needed to support them in performing the safe, orderly, and efficient management of flights. On the CWP, flights are presented as radar tracks with an associated label indicating as a minimum the flight identification or callsign, current flight level, current speed, and the next point of the route.

While performing their tasks, ATCos must communicate with flight crews to provide them with commands and information. This communication can be performed via voice or via datalink.

Communication between ATCos and FC follows the standard established by the International Civil Aviation Organization (ICAO) [1]. This standard states that when communications are initiated by ATCos they must:

- Start by the identification or the callsign of the flight being addressed;
- Continue by issuing the command with its qualifiers or information.

Example:
Iberia three four two descend flight level two five zero

Control commands safety-related parts must always be acknowledged by the FC whose answer:

- Starts with the command with its qualifiers;
- Ends with the identification or callsign of flight.

This answer is known as readback and it is vital for ensuring mutual understanding between the FC and the ATCo of the intended plan for the aircraft. ICAO [2] requires "Flight Crew shall read back to the air traffic controller safety-related parts of ATCo clea-rances and instructions which are transmitted by voice".

The answer to the previous command would be:

Descending to flight level two five zero, Iberia three four two

When the FC initiates communication with the ATCos they will start the communication with the callsign and follow it with the necessary information. FC can initiate communications for several reasons:

- FCs always have to call air traffic control when they are about to enter a new air traffic service, ATS, unit or sector; they make a call prior to the boundary between both airspaces. The FC communicates with the ATCo to make them aware of their presence and confirm that voice communication is feasible for emergency use. In this communication the FC will typically greet the ATCo and provide some information related to the flight. Example: Good morning Ryanair nine zero three five flight level three hundred.
- The FC usually starts communications at any time with ATCos to request modifying vertical/horizontal trajectories and/or the speed to fly at the optimum performance of the aircraft.
- Another important reason to initiate a call from the FC is requesting to modify their flight level, route, speed, or any other flight condition because of adverse weather such as encountering cumulonimbus, severe turbulence, icing etc. Example: Air Europa six alfa bravo requesting flight level four zero zero due to severe turbulence.

The ATM community has investigated ASR mainly using communications from controller utterances [3–5]. This is due mainly to the fact that the ASR is seen as a means to free the controller from the necessity to manually introduce commands on the CWP, but also because of the characteristics of controllers and pilot communications.

There are some basic features in communications initiated by the controller:

- The voice signal used for speech recognition from controllers' voice utterances is extracted directly from the jack of the controller. This signal has a low degree of noise.
- Controllers' language is English or the local language of the ground station [6].
- Usually, controllers of an air navigation service provider will have similar accents when speaking.
- The percentage of women/men in air traffic control differs from one country to another. In Spain or France, the percentage is around 33% women [7,8].
- Communications from controller to flight crew can be standardised as [9]: call id + command + qualifier 1 + qualifier 2.

On the other hand, there are some basic features in communications initiated by flight crew.

- The voice signal used for speech recognition from flight crew voice utterances is extracted from radio communications. The quality of these communications is highly dependent on:
  1. The distance of the aircraft to the receiving radio station.
  2. The signal-to-noise ratio, SNR, can vary from 10 dB to −5 dB [10].
  3. The quality of the signal transportation from the radio station to the air traffic control facility where the signal is analysed.
- Flight crew language is English or the local language of the ground station [6].

- FCs have very different accents usually, but not always, relating to the flight company country. Countries that are in the routes of international flights have even higher rates of different accents.
- Communications from flight crew to controller can, similarly to the controller's ones, be decomposed as: call id + command + qualifier 1 + qualifier 2. Alternatively, if it is a readback: command + qualifier 1 + qualifier 2 + call id.

Finally, there is environmental information that can be exploited. Each ATCo has a list of flights that either are in their sector, are about to enter into it or are of interest (e.g., because they fly near the sector border). This information is provided by a flight data processor (FDP) that ensures that the list of flights is updated with new incorporations and cancellations once the flight is no longer of interest.

The level of automation is having continuous improvements and enhancements introducing new functions to assist the ATCo for better situational awareness and a reduction in workload supporting them to focus attention when and where needed. Within these new functions it is the ASR Project that requires a new Human–Machine Interface (HMI) presentation. The new methods of interaction have to be compatible with the other systems and subsystems within the CWP to benefit the controller's duties.

Identification of the key information present in the communication exchange is necessary to provide the new HMI. Information extraction from written text can follow very different approaches and several factors (language, domain, entity type) impact the selected technique [11,12]. The extraction of information in the ATM domain has mainly used knowledge-based methods and machine learning models [5,13].

The work performed in the project uses as baseline an ASR prototype that has been developed between Enaire, Crida and EML to support the quantification of the controller's workload [14,15]. The prototype follows a hybrid architecture and uses a knowledge-based method to identify the callsign. The performance of the algorithm was considered adequate in its previous use, thus the latest investigations in information extraction have not been considered in this work.

## 2. Materials and Methods

As presented in the introduction, the presence of the flight identification or callsign is a common feature in the communications procedures in current operations. Callsign recognition and illumination is considered as a quick win by Enaire that can be implemented in any CWP as they are equipped with a radar surveillance service that can display the radar track and callsign identification regardless of the unit where they are installed: en-route/terminal-manoeuvring area, TMA, or in tower, TWR, units. The work hypotheses are:

**Hypothesis 1 (H1).** *The integration of an ASR system in an operational CWP and voice communication system, VCS, can be performed without negatively impacting the other system.*

**Hypothesis 2 (H2).** *ASR will decrease ATCos' workload by guiding the attention of the controller to the aircraft demanding an action.*

**Hypothesis 3 (H3).** *ASR will increase of ATCos' situational awareness by quickly identifying new flights entering the sector or flight crews requesting actions from ATCos.*

**Hypothesis 4 (H4).** *ASR will increase aviation safety by illuminating the callsign coming from an ATCo's utterances ensuring they are addressing the proper aircraft.*

To test the hypothesis a prototype that meets several ATM related requirements was developed and later tested through two different and complementary approaches: a real-time simulation and a statistical analysis. Real-time simulation is a human-in-the-loop technique that tests a concept in a controlled, repeatable, and realistic environment. The technique is well established in the air traffic management environment to validate tools

and concepts [16,17]. It provides qualitative and quantitative feedback. Some aspects of the technique that need to be taken into account are the level of realism of the environment, the adequacy of the test subjects, and the number of runs that takes place.

The real-time simulation was performed on Crida's premises with Enaire controllers, and a prototype developed by Indra the ASR engine provided by "EML Speech Processing Server" in November 2021. The environment realism was high as operational controller working positions and voice communication systems were used as hardware while the simulation scenarios were based on Spanish operational scenarios. The test-subjects were Enaire's operational controllers with over 10 years of experience, supported by professional pilots acting as pseudopilots. The number of runs was low as only six runs were performed.

The statistical analysis was designed to address two weaknesses of the real-time simulation, RTS. One of the weaknesses is the difference between ATCo–Pilot communications in a laboratory and operational environments. The second one is the low statistical feedback due to the low number of runs. The statistical analysis uses operational recordings from Spanish airspace. The analysis took place in February–March 2022.

### 2.1. Requirements to Be Met by the System

To perform the experiment several requirements have been identified on the ASR engine and on the voice recognition system, VRS.

### 2.1.1. Basic ASR Engine Requirements for Callsign Identification

Due to the ATM application in which voice recognition is going to be used, there are three outstanding requirements:

- The voice recognition system, VRS, shall be able to function without connection to sources external to the area control centre, ACC.
- The callsign illumination must be produced as soon as possible once the communication has started.
- The ASR engine shall be able to process the utterance in English and the local language, when local languages are allowed.

The first requirement limits the available ASR engines, as it must be autonomous. Enaire considers flight management as a strategic field, and therefore, an ACC must be able to provide its service even if it is isolated. This is a requirement set by Enaire that may not be shared by other air navigation service providers, (ANSP). This requirement may nevertheless change in the near future to align with the strategy to deliver the European commission's Digital European Sky [18] and Enaire's strategic plan [19].

The second requirement implies that the ASR engine must be able to perform in streaming and provide partial transcriptions. As in most of the use cases, the callsign is at the beginning of the phrase and the time of the initiation of these partial transcriptions is also critical. In project PJ.10-W2-96 ASR [20,21], the requirement has been established at one second after the ATCo has said the callsign. This value needs to be validated.

### 2.1.2. VRS Requirements for Callsign Identification

The fact that a mistake in callsign illumination can mislead the ATCo which in turn may provoke an accident puts in place a new set of requirements on the VRS:

- It is preferable not to have a callsign illumination rather than a wrong callsign illumination.
- The VRS will use the sector flight list from the CWP to improve its performance.

The callsign detection algorithm also should include the callsign rules defined by ICAO [1] and is able to detect a flight indicative independently of the method or the language used by the controller (pilot) to address it. These methods include:

- The radio callsign e.g., Beeline/Cactus.
- The company name e.g., Brussels Airlines/US Airways.
- ICAO designator using aeronautical alphabet. e.g., Bravo Echo Lima (BEL)/Alpha Whiskey Eco (AWE).

- All of the possible modes to pronounce a number. e.g., one zero zero, ten zero, one hundred.

ATCos may pronounce more than one callsign in one utterance, e.g., because they give instructions to different aircraft or because they are informing a flight about a traffic that may influence them.

ATCos and FCs are allowed to refer only to partial callsigns once the first communication is established and there is no possibility of confusion.

### 2.2. Real-Time Simulation

To investigate the benefits of callsign illumination through a real-time simulation, a VRS has been integrated in an Enaire's operational CWP. The VRS was developed by Indra based on Voice, a recognition system developed by Crida using EML's ASR engines. The ASR models contained have been developed jointly by EML and Crida [14] and are able to work on-the-fly processing the audio signal in real-time streaming.

Enaire's ATM system is SACTA (Air Traffic Control Automatic System) developed by Indra. The communication system that processes audio signals has recently been upgraded to COMETA (Integrated Voice IP Communication into SACTA). Figure 1 presents the architecture used. The audio is extracted by the audio extractor and sent to Voice for speech and event recognition. The delivery module then sends the event (callsign highlight) information to the SACTA CWP. The SACTA CWP also sends the environmental information to Voice via the delivery module.

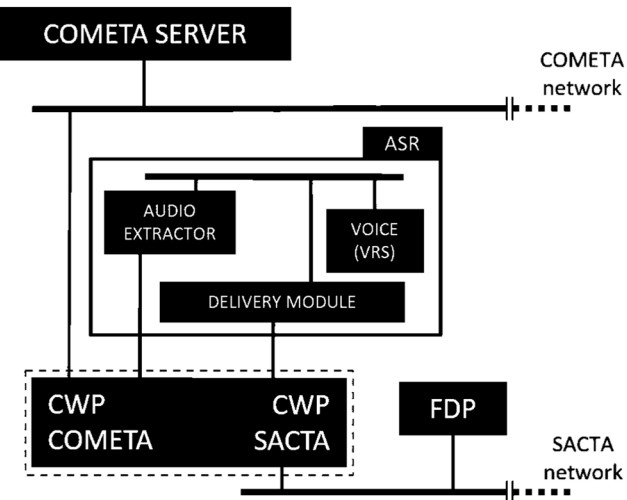

**Figure 1.** System Architecture.

The VRS module uses environment information to improve the recognition rate and allows the system to perform a safety check on the correct identification of the flight.

COMETA processes the audio signal following the aeronautical standard [20]. The raw audio is extracted and provided to the VRS. COMETA distinguishes between controller and FC communications. The signal is tagged with a flag indicating the source, FC (0) or ATCo (1), Figure 2. The ATCo can be in charge of one or several frequencies for radio reception (RX) and transmission (TX) depending on the sector configuration, E.g., one planner controller may listen not only to the frequency of their sector, but also to the frequency of the adjacent TMA sector to increase the situational awareness of departing flights. The system is linked to the frequency that the controller transmits (TX).

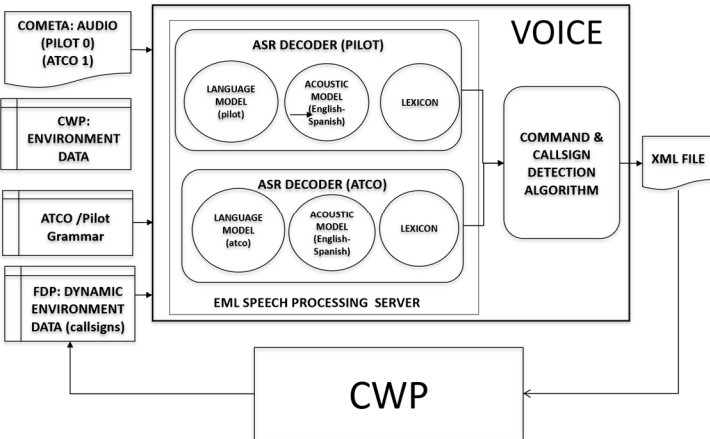

**Figure 2.** VRS Architecture.

The list of possible flights of interest can be provided to the VRS from two different sources.

- The FDP has the list of flights that are of interest for the ACC (composed of several sectors). The FDP ensures that the list of flights in each CWP is updated with new incorporations or cancellations once the flight is no longer of interest.
- The CWP has the list of flights that are of interest for the sector. This list is smaller than the previous one, but some flights may not be covered, for example, last minute flights deviated due to weather.

After performing a cost/benefit analysis regarding the size of each file, the system implications, and the number of flights that may be impacted, Enaire decided that the CWP would be the one providing the list. This list is provided to the dynamic lexicon update feature of "EML Speech Processing Server" (to enhance the callsign detection) and to the detection algorithm. It is updated dynamically.

ASR is provided to the "Voice" application with the "EML Speech Processing Server". The recognition engines are set for real-time streaming transcriptions with partial results, using the latest state-of-the-art technologies in ASR such as bidirectional long-short term memory (BILSTM) neural networks [22] for acoustic modelling, voice activity detection (VAD) [23] and a dynamic lexicon update feature. Due to the different characteristics of the communications, two different recognition models have been developed. Both models use the same multilingual (English and Spanish) acoustic model trained with 1000 h of recordings (about 400 h of English and 600 h of Spanish recordings out of the ATM domain) and then adapted with approximately 100 h of ATM domain recordings, which were manually transcribed from operational controller communications (these contain phrases that can be in Spanish, English, or mixing both). The difference between both recognition models is on the two class-based language models, one developed for controller communications (ATCo), and one for flight crew communications (FC). The ATCo language model is a more mature model that has been trained with operational transcriptions (approximately 90 k) gathered along several years of collaboration between Crida and EML. The FC model is a newer model that has been adapted with approximately 14 k transcriptions (from only 12 h of flight crew operational recordings).

The development of the Spanish–English acoustic model addresses some of the challenges associated with multilingual speech [24]: the simultaneous use of several languages in one sentence and the different speech rates related to each one.

The "EML Speech Processing Server" dynamic lexicon update feature enables language model class entries to be defined at runtime without having to restart the recognition engines, allowing the "Voice" application to update for a given use. The waypoint class allows the use of the same language model for different ACCs by only changing the list of entries for the waypoint class. The callsign class allows to quickly adapt the model to a specific sector, date, and time by providing the ATM-planned flights.

The command and callsign detection algorithm analyses the text recognised by the "EML Speech Processing Server" and classifies the words according to the most probable value. The algorithm follows a rule-based grammar approach [25], that was developed to support the analysis of the controller's workload calculation through a postprocessing method that included several different sources of information including flight plans, radar tracks and radio communications [14,26]. It not only classifies callsigns but also the different type of commands that are issued by controllers such as flight level or speed change. The algorithm was updated to take into account the information provided by the flight plans.

ICAO annex 10 [6] rules for callsigns are used within the algorithm to identify a possible callsign. These rules indicate that callsigns have three letters to identify the aircraft operator, followed by between one and four alphanumeric characters. The algorithm also includes the requirements listed in the VSR requirements for callsign identification paragraph. Another input to the algorithm is a list of keywords that provides possible callsign identification (radio callsign, company name, and ICAO designator) of the companies that are (or have been) authorised to operate in Spain. Finally, the algorithm increases or decreases the probability of callsign taking into account the language model; as an example, in controllers' utterances the callsign is usually at the beginning of the phrase and is followed by a command.

Once a sequence is classified as a probable callsign, it is compared against the list of possible callsigns received from the FDP. This list has the identifications of the flights that are in or near the sector and are of interest to the controller. In the implementation performed, only complete callsigns are recognised: only when the complete set of alphanumeric characters has been completely transcribed, identified by the callsign detection algorithm and are present in the FDP list, the VRS considers that there is a match. When a callsign is positively detected, a file with the information is created and sent to the Human–Machine Interface, HMI. The HMI displays a white circle around the radar track that flashes during a configurable time, currently set as 5 s, see Figure 3. The HMI is able to highlight up to 5 callsigns simultaneously, as more than one communication can be performed during this time.

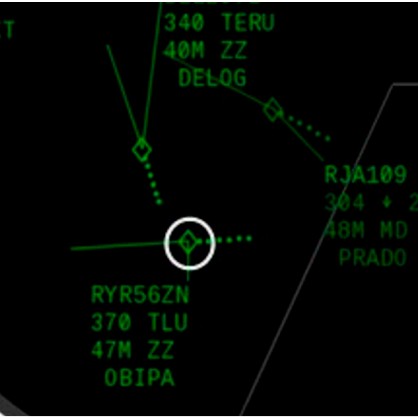

**Figure 3.** Radar track highlight following callsign detection (Captured from Supplementary Materials).

The real-time simulation addressed two sectors of the Madrid Flight Information Region, FIR, which has medium complexity. The sectors are Zamora–Toledo Integrated, LECMZTI, in blue in Figure 4, and Castejon–Zaragoza Integrated, LECMCZI, in red. The figure has been obtained from Enaire's Aeronautical Information web application [27].

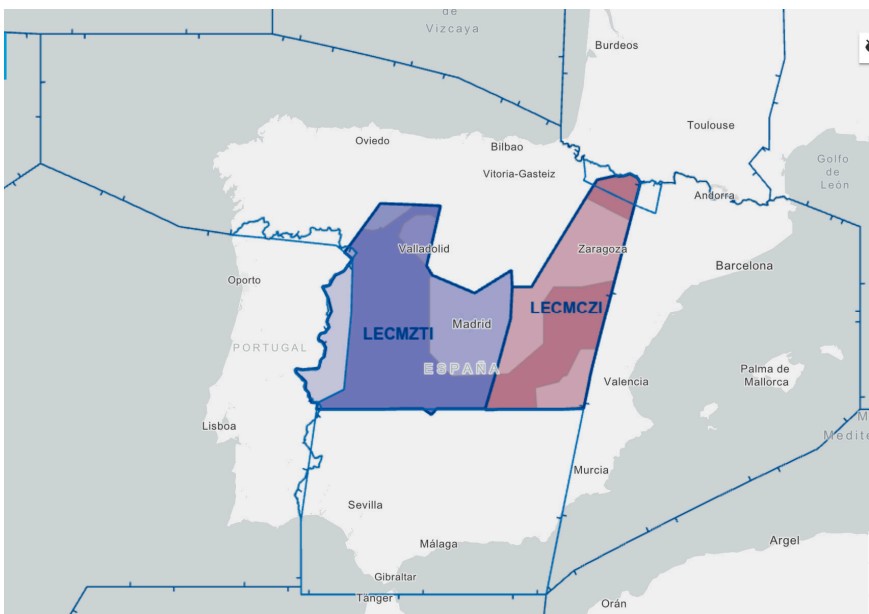

**Figure 4.** Simulated sectors in the real-time simulation.

This configuration is an operational configuration that is used at night. This configuration facilitates the evaluation of the validation objectives:

- The sectors have several entry points where the flight crew performs their first call (related to the highlight of callsigns on the CWP from pilot utterances).
- The sectors are quite wide and integrate nine control volumes. This implies that there are very different traffic flows that require different types of control and facilitates the creation of situations where the traffic is focused in one area or disperse along the whole sector (related to both, the highlight of callsigns on the CWP from the pilot and controller's utterances).
- There are several airports within the control volume, the main one being Madrid- Barajas airport, LEMD in its ICAO code. This airport was used in the north configuration and generated traffic flows to/from both sectors.

Controllers were operational controllers from Madrid FIR; thus, they were familiar with the scenario and the control rules. The control operation rules used were the operational ones with one simplification: the lower level to hand over traffic to TMAs and airport in all the volumes was the same, FL210.

Two types of exercise were used. Both had from medium-to-high traffic loads that supported the test of technical and operational requirements. The traffic for the exercise was created by adapting real traffic from 14 July 2019. The traffic adaptation included the modification of callsigns, flight levels and entry times. The traffic sample covered 67 different airlines plus 10 general aviation registration numbers.

The exercises were performed in an integrated controller position: one controller performs the executive and planning roles. One pseudopilot was assigned to each position. The pseudopilots have an active pilot license and have participated in previous validation activities.

The first exercise had a constant flow of aircraft that entered the controlled sectors from collateral dependencies and adjacent sectors. There were traffic peaks to concatenate calls and facilitate situations where the controller focused on one part of the sector. The traffic flight followed instrumental flight rules, IFR. There were flights from commercial airlines and general aviation.

The second exercise started in one sector configuration (one controller in charge of both sectors, Config R21 is used at nights with low traffic). The traffic steadily increases and the exercise leader, acting as supervisor, decides to split the sectors around minute 10. The traffic continues increasing until it starts to decrease. Near the end of the exercise, the traffic

is low again and both sectors are grouped again. The exercise ends with one controller managing both sectors. The exercise allows the analysis of the requirements related to sectors splitting and grouping, the rapid successive communications from different pilots, and the overlap of communication from different aircraft. Traffic followed instrument flight rules (IFR), visual flight rules (VFR) and operational air traffic (OAT).

The simulation took place across two days. Each day controllers performed three runs, one with the reference scenario (without the ASR) and two with the solution one. Controllers rotated between the different sectors in each run. The results were gathered by data logs, questionnaires, debriefings, and observations.

### 2.3. Statistical Approach

As the RTS had some limitations, i.e., the number of scheduled runs was low, the number of controllers and pseudopilots was limited to two of each, and finally, the utterances that would be analysed were from a simulation environment which could impact the natural language of the speakers. To overcome these limitations a statistical approach was planned. The statistical approach includes the analysis of operational recordings from different types of sector and several actors, both controllers and flight crew.

The statistical test was performed between January and February 2022 with recordings from 2019. It was decided to use recordings from 2019 due to the impact of the COVID-19 pandemic on the amount and diversity of flights between 2020 and 2021.

Operational recordings fed the Voice prototype to obtain information on the callsign and event identification. The outcomes from the prototype were compared to a gold standard manually created, and rates regarding callsign and command identification have been obtained through comparison.

From the architectural point of view, this approach used part of the previous RTS prototype. It used the "Voice" prototype and callsign algorithm to transform the audios to text and identify the callsigns. Flight plans and environment data were also provided, although not dynamically, and finally the audios did not come from the COMETA system as it was not deployed at the time.

The operational environment for the statistical approach belongs to the sectors Lower Castejon (CJL), Upper Castejon (CJU) and Santiago (SAN), all of them from Madrid ACC (LECM). CJL and CJU were also used in the RTS. Figure 5 [27], presents the locations of the sectors within Spain's airspace, SAN in light blue and CJL and CJU in dark blue.

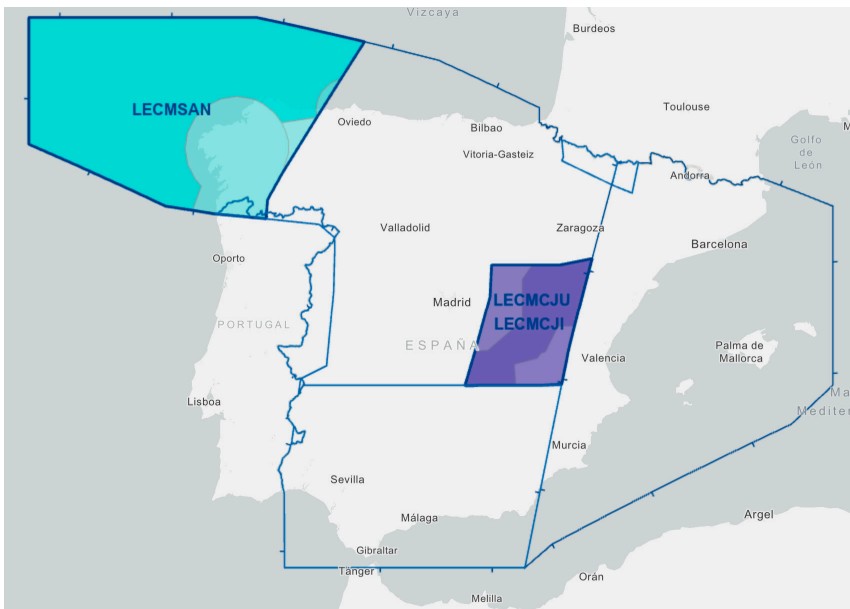

**Figure 5.** Simulated sectors in the statistical approach.

These sectors were selected due to their complementary characteristics that provide a wide sample of technical (i.e., signal-to-noise ratio, native speakers' origin) and operational (i.e., type of commands) characteristics:

- CJL is a sector with good radio coverage whose main traffic flows are to and from Madrid-Barajas Airport, the major Spanish airport. It limits with Madrid TMA and the surface. Control service is provided to all aircraft from FL210 to FL325. Information service is provided from SFC to FL210 outside the TMA/airport areas and airways.
- CJU is a sector with good radio coverage and quality whose main traffic flows are to and from Madrid-Barajas Airport, and over flights to the south of Spain. Control service is provided to all aircraft from FL325 to FL660.
- The SAN sector includes a large proportion of oceanic airspace that has lower radio coverage compared to CJL and CJU. Its main flows are overflights to/from the America, and to/ from United Kingdom. Free route airspace is implemented in this sector. Control service is provided to all aircraft from FL210 to FL660. Information service is provided from SFC to FL210 outside the TMA/airport and control areas.

## 3. Results

Technical results were gathered from system and data logs, while operational results have been collected from questionnaires, observations, and debriefings.

### 3.1. Technical Results

The real-time simulation produced 1139 communications from ATCo and FC. Several of the recordings were disregarded in the final analysis because they were just noise or did not contain a callsign. The traffic sample covered 67 different airlines plus 10 general aviation registration numbers that were addressed either in Spanish or English according to the airline origin.

Callsign recognition rates obtained from the RTS analysis appear in Table 1. The row speaker indicates the number of callsigns contained and detected in controllers' or flight crew utterances. No false recognition was performed.

**Table 1.** Callsign recognition from RTS analysis.

| Speaker | Callsigns | Detected Call Sings | Detection Rate |
|---|---|---|---|
| Controller | 859 | 721 | 84% |
| Flight Crew | 687 | 457 | 67% |

Regarding the identification of numbers within callsigns:

- Thousands are correctly identified (100%);
- Hundreds are correctly identified (100%);
- Numbers between 11 and 99 (e.g., 13, 18, 34) have very high recognition rates (98%);
- Numbers with triple have different success rates;
  - 111 (triple one) was transcribed correctly 88% and one time as 341;
  - 666 (triple six) had 67% success but was transcribed 326;
  - 777 (triple seven) was the least accurate of the group being transcribed as 37 in 84% of the utterances;
  - 888 (triple eight) has a success of 75%, but was transcribed as 68 two times.

One problem detected during the analysis is that if the transcription misses the ICAO name of the company, the algorithm may identify groups of four numbers and letters as possible callsigns. Due to the flight list check performed these wrong identifications were not presented to the CWP.

The statistical approach used 449 operational recordings from ATCo and FC utterances. Several of the recordings were disregarded in the final analysis because they were just

noise or did not contain a callsign. The traffic sample covered 29 different airlines from 18 different countries.

Callsign recognition rates obtained from the analysis appear in Table 2. No false recognition was performed. An additional study was performed in the statistical approach. The row first call/request in the table is a subset of the pilot utterances where the FC initiates the communication, i.e., the first time a flight enters a sector or a request from the pilot not expected by the controller. These communications are of special interest as they imply a change to the controller's attention focus.

**Table 2.** Callsign recognition from statistical analysis.

| Speaker | Callsigns | Detected Call Sings | Detection Rate |
|---|---|---|---|
| Controller | 143 | 127 | 87% |
| Flight Crew | 158 | 77 | 49% |
| Flight Crew First call/request | 65 | 38 | 58.5% |

In the statistical analysis a review taking into account the airline was also performed.

The percentage of correctly detected callsigns is higher for ATCOs than for flight crew in both cases as the algorithm is optimised for the ATCo utterances.

Regarding the comparison between simulation and operational recordings, the percentage of the ATCos are similar but the percentage of the flight crew is better in the simulation. This was already expected as the quality of the recording (ratio signal-to-noise) is better in the simulation and the accent (mother tongue) of the pseudopilots is unique (Spanish) while the one from the operational recordings is very diverse with companies from 18 different countries.

No callsign was wrongly recognised as only complete callsigns were detected. Feedback from the controllers indicated that they would like to have higher recognition rates even if some callsigns were incorrectly detected and highlighted.

In the statistical analysis detection per company was also performed. Table 3 presents the analysis. In the table the companies with less than five appearances have been removed as they have been considered that the sample is too low to infer a tendency.

**Table 3.** Callsign detection per company and speaker.

| Airline | | Controller Utterances | | | Flight Crew | | |
|---|---|---|---|---|---|---|---|
| | | Call Sings | Detected | Rate | Call Sings | Detected | Rate |
| American Airlines | AAL | 5 | 4 | 80% | 4 | 2 | 50% |
| Air Europa | AEA | 8 | 8 | 100% | 7 | 6 | 86% |
| Aegean airlines | AEE | 9 | 6 | 67% | 6 | 0 | 0% |
| Air Nostrum | ANE | 5 | 5 | 100% | 5 | 5 | 100% |
| Condor | CFG | 9 | 8 | 89% | 8 | 3 | 38% |
| Iberia | IBE | 8 | 8 | 100% | 11 | 11 | 100% |
| Iberia Express | IBS | 7 | 7 | 100% | 4 | 1 | 25% |
| Ryanair | RYR | 18 | 17 | 94% | 23 | 11 | 48% |
| Tap Portugal | TAP | 7 | 7 | 100% | 11 | 4 | 36% |
| Thomson | TOM | 14 | 13 | 93% | 18 | 12 | 67% |
| Emirates Airlines | UAE | 11 | 11 | 100% | 14 | 4 | 29% |

From the analysis of the utterance according to the company, it can be inferred that in controllers' utterances, most of the airlines have over a 90% detection rate except Condor (89%), American Airlines (80%), and Aegean Airlines (67%).

In flight crew utterances the detection rate highly varies from one company to another. Spanish companies have high detection rates, i.e., Air Nostrum (100%), while for other companies the detection varies from 67% (Thomson) to 29% (Emirates), with the notable exception of Aegean Airlines that is not recognised anytime.

The callsign illumination took 3.02 s after the initialization of the utterance to the illumination in the CWP during the RTS. If the callsign was at the end of the phrase the reaction time of the prototype was lower, 0.93 s.

### 3.2. Operational Results

Operational feedback was obtained in the RTS regarding human performance (workload, accuracy, timeliness, and coherency of the information provided) and safety (situational awareness and errors induced by the prototype).

### 3.2.1. Human Performance

Workload was collected through Nasa-TLX [28], tailor-made questionnaires, and debriefings. The Nasa-TLX scored 9.1 (out of 20) for the reference and 7.9 (out of 20) for the solution questionnaire. The tailor-made questionnaire and debriefings indicated that workload slightly decreased in the solution scenario.

Accuracy was collected through tailor-made questionnaires, debriefings, and data logs. The feedback was that the tool needed improvement in the recognition rates to be able to effectively support them. Controllers indicated that they would prefer some occasional false positive callsign recognised if that would mean higher recognition rates.

Timeliness was collected through tailor-made questionnaires, debriefings, and data logs. The timeliness rated as adequate for the callsigns at the end of the utterance but inadequate when the callsign was at the begging of the utterance.

Feedback on the coherency of the information provided was collected through tailor-made questionnaires, and debriefings. Controllers were satisfied with the flexibility of the tool that allowed them to address the flight using very different approaches i.e., using English or Spanish, the radio name, spelling, numbers in hundreds, thousands, double, or triple. They also appreciated the HMI of the symbol on the radar track, its font, colour, and duration. A request was made to make the HMI different to be able to distinguish when the callsign highlight was from controller or flight crew utterances.

### 3.2.2. Safety

Situational awareness was collected through SASHA questionnaires [29] and debriefings. The overall score of the SASHA questionnaire was 4.0 (out of 6) in the reference questionnaire and 4.4 (out of 6) in the solution questionnaire. The situational awareness improved slightly with the use of VRS. During the debriefings and tailor-made questionnaires, controllers stated that situational awareness was improved but they considered that the VRS recognition rates were not high enough to allow them to completely confide and exploit the tool. They consider that higher callsign recognition rates and timeliness would further improve their situational awareness.

Errors induced by the prototype were collected through debriefings and data logs. No error resulted from the introduction of the VRS. No false recognition was performed during the simulation. This can be attributed to the requirement that indicates that only callsigns in the FDP list with the complete alphanumeric sequence provide a positive detection.

### 3.2.3. Additional Findings

Finally, when asked by each individual functionality, controllers appreciated especially the identification of flights from flight crew utterances. They considered that with higher robustness (meaning accuracy and timeliness) it would support to develop their tasks

more efficiently, reduce workload and increase situational awareness. They considered that identification of flights from controller utterances could be especially helpful when a controller needs to understand the sector situation but is not located directly in front of the screen. On these occasions, following the performance of the controller on the radio can be difficult and having a callsign highlighting the flights from controllers' utterance will support them. At Enaire this situation happens:

- During a shift change. The entering controller may sit near the departing controller during a period of time to be able to grasp the situation before actually controlling the flights.
- When new controllers have onsite training. The new controller may be near the experienced controller following the issued commands, or a supervisor may be near the new controller.

## 4. Discussion

The experiment was able to connect a preindustrial VRS prototype with an ASR engine with operational systems. This connection included an operational SACTA 4 CWP that provides context information in real time to the VRS (flight plan list in this approach), receives information from the VRS and presents it to the controller in a coherent approach with the rest of the CWP information. The exercise also included connection with an operational voice communication system, COMETA, that provided the ATCo–Flight Crew communication exchange following the aeronautical standards. This integration was performed without any impact to either of the systems, demonstrating the feasibility of the technical solution. It is especially significant that no delay was introduced in the voice exchange between both actors. Hypothesis 1 is confirmed.

Although the callsign recognition was not as high as in other controller speech recognition studies [30,31], controllers' workload was reduced, and situational awareness was increased. Hypotheses 2 and 3 are confirmed. This outcome aligns with other studies [32], about the importance on high callsign recognition rates on the ATCo's perception of ASR technology support. The improvement in workload and situational awareness was not as high as expected due to the accuracy and timeliness of the VRS prototype.

The controller model has higher recognition rates than the FC model. Callsign recognition in FC utterances from the RTS was higher than from the operational ones. These outcomes were already expected due to the different maturity of both voice models; the inherent aspects of operational FC utterances with worse signal/noise relation, and higher number of different accents when compared with the controller model.

Callsigns in Spanish or with a Spanish accent have higher recognition rates than the rest. This outcome is attributable to the callsign training database which is composed mainly of Spanish controllers' utterances. The recent update with FC utterances has not been enough to cover the gap.

The bad results of some companies can be related to two different causes, training of the ASR model and phonetization of the company.

A low representation of the company in the training database would reflect in both the callsign identification from controller and flight crew utterances. Two examples of this effect can be identified in Aegean airlines in the statistical approach (67% ATCo recognition rate and 0% FC recognition rate), and Nile Air airline (Nile Bird or NIA callsign) in the RTS execution (0% FC recognition rate). Regarding this last example, Nile Air, it should be noted that, during the RTS, controllers spelt the callsign (NIA) as they were not familiarised with the airline. By doing this, they obtained a 100% recognition rate.

An incorrect or incomplete phonetization of the airline callsign would also provide low recognition rates. The airlines' callsigns are phonetised in English and Spanish, and enriched with the training database. One example of this problem is the airline Atlantic Airways whose callsign is Faroeline. Faroeline is correctly transcribed and identified in Spanish utterances but was transcribed as Flyer nine in pilot utterances. The lack of training could be enriched by enlarging the possible phonetisation with the native language of the airline country. Nevertheless, this method has its drawbacks as depending on the

airline company, the nationality of pilots can be very diverse. As an example, Ryanair has a multinational group of pilots which, in 2019, counted with people from 53 different nations [32].

Future implementations where controller utterances are used to automatically implement the command on the CWP will require the recognition of partial callsigns, as already performed in some experiments [33]. This approach requests the review of the algorithm to minimise the identified problem of grouping alphanumeric characters not related to callsigns. The trade-off between false positives and recognition rate is something that needs to be investigated.

The FC usually speak differently when they start the dialogue. The FC wants to grab the attention of ATCos and are conscious that the controllers need to change the attention focus. Therefore, when initiating contact or check-in calls, they usually speak louder, slower and pronounce more clearly. The identification of these flights was one of the most appreciated by controllers, which supports the feedback provided in other ATC speech recognition experiments [34].

As already mentioned, other studies have higher callsign recognition rates [30,35]. These studies use a statistical approach for the information extraction, such as machine learning algorithms or deep neuronal networks. Comparison between both methods indicate the rule-based algorithm outperforms the statistical approach [36]. It should be noted that the statistical approach method is very dependent on the training database [35]. The method used, rule-based grammar, enhanced with the improvements identified in database training and company phonetisation, should be compared with a statistical approach that takes advantage of the Enaire's database.

Regarding the timeliness, the difference in the recognition time depending on the position of the callsigns in the utterance impacts greatly the support perceived by the controller. Further investigation in the ASR engine, ASR models, and lexicon is being conducted in order to reduce the time of partial results sent by the "EML Speech Processing Server" to then be processed by the "Command & callsign detection algorithm" within the "Voice" application.

Although situational awareness improved and no error was induced by the VRS, controllers did not consider the use of the identification of the callsign from a controller utterance as a safety tool. They considered this use of VRS as helpful to provide context information to other controllers. Hypothesis 4 is rejected.

As the feasibility of the integration of the VRS system has been demonstrated and the identification of callsigns from FC utterances provides benefits in terms of workload and situational awareness, the way forward is the improvement of recognition rates and timelines at the beginning of the utterance following the approaches previously identified. The identification of callsigns from controller utterances is also of interest to support controllers in handover and on-hand training. Improvements in recognition rates and timeliness are less critical in this case, although also necessary.

**Supplementary Materials:** The following supporting information can be downloaded at: Video: https://www.youtube.com/watch?v=qLQVoS4Qkms (accessed on 28 February 2023).

**Author Contributions:** R.G.: conceptualization, formal analysis, methodology, investigation, validation, writing—original draft preparation, visualization; A.F. and F.C.: data curation, formal analyses, methodology, software, writing—review and editing; J.A.: investigation, methodology, validation, writing—original draft preparation; C.P.d.O.: methodology, software, writing—original draft preparation; C.B. project administration, conceptualization, supervision, resources, writing—review and editing. All authors have read and agreed to the published version of the manuscript.

**Funding:** This research was funded by SESAR Joint Undertaking under the European Union's Horizon 2020 research and innovation programme under grant agreement No. 874464.

**Data Availability Statement:** Data are not available for public consultation following Spanish law for Air Traffic Regulations, BOE-A-2018-15406 and General telecommunications regarding controller–pilot communication exchange.

**Acknowledgments:** To Nadal Ceñal, Julian Chaves and Mhamed Fillal from Indra that contributed to the prototype development, integration, supported the exercises and reviewed this article.

**Conflicts of Interest:** The authors declare no conflict of interest.

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
