# Peer review of "Automatic Flight Callsign Identification on a Controller Working Position: Real-Time Simulation and Analysis of Operational Recordings"

_aerospace, doi:10.3390/aerospace10050433_

Round 1
Reviewer 1 Report
For improving the recognition rate of callsigns, this paper utilizes the phonetization of airline designators. In addition, by considering the partial ASR hypothesis situational awareness improved and a reduction in the controller's workload was observed. Workload reduction and improving the callsign recognition accuracy are part of the research interest in this community. However, the applied methods are moderately novel. This is interesting to compare the performance of the proposed method with the sequence modelling approaches for callsign recognition e.g., BERT-based name entity recognition (NER) for callsign extraction. In addition, this is better to use the recent end-to-end (E2E) ASR models rather than the BILSTM structure. In addition to the system architecture, this is better to illustrate the proposed method. There are many typos in this paper that needs to be reviewed.
Reviewer 2 Report
The paper presents a system to use ASR to reduce ATC workload. The ASR used was jointly developed by the authors, but details of what ASR was used is limited! What architecture of ASR was used -- hybrid or end-to-end? How was it trained -- CTC, cross-entropy, attention-loss, or something else? Even if the goal of the paper is to show the benefit of automating some tasks with machine learning technologies, such crucial details should be included. If the ASR was trained with multilingual data, what languages were used? What is the distribution of these languages in terms of amount of training data? Even the performance of ASR doesn't seem to be evaluated.
The language model is supposed to be "mature", but it is hard to understand what is meant by that. It is mentioned that class-based LM was used, but please provide a reference.
On the other hand, I find that the discussion in Sections 3.1 and 3.3 quite interesting. Even negative results such as 0% detection rate are interesting enough if simple examples of how the system is failing is provided, which might be useful for the readers to learn and understand.
Author Response
Thank you for the review. Please see the attachment.
Regards

Reviewer 3 Report
The introduction describes well the problem of controller-pilot communication but does not deal with the detection of semantic entities at all.
Insufficient description of ASR and callsign detection.
The call sign detection is based on the input text. The use of the recognizer lattice is very beneficial. How is classification/detection done from the text?
Text formatting is inconsistent, eg sections 2 and 3.
It is not explained why there were no callsign detection false alarms. Possible due to filtering the output according to existing callsigns during the test?
It's not well explained why callsign highlighting is 3x faster for FC. Does ASR work in full utterance recognition mode or on-the-fly mode? Wouldn't it be better to use a keyword detection or keyphrase detection approach?
Minor suggestion:
- line 53: missing end dot
- line 90-106: mixed bullet points and a numbered list (indent numbered part?)
- wrong hyphenation: an-other, standard-ized, hypoth-eses, ...
- confusing text formatting: 164-171
- typos: established, locations , ..., ENAIRE vs Enaire, in Plan de Vuelo 2025 [1415], (pseudo)pilots was, appear on Table 1/2, tailor made vs tailor-made
- separating the description of the table 1 and the table itself.
Round 2
Reviewer 1 Report
Some points about the novelty of the proposed methed explained. This manuscript can be improved by revision and comparing with baselines for callsign extraction.
Author Response
Thank you for your time and your review. Please see the attachement.
